# The Influence of Polyphenol Compounds on Human Gastrointestinal Tract Microbiota

**DOI:** 10.3390/nu12020350

**Published:** 2020-01-29

**Authors:** Michał Wiciński, Jakub Gębalski, Ewelina Mazurek, Marta Podhorecka, Maciej Śniegocki, Paweł Szychta, Ewelina Sawicka, Bartosz Malinowski

**Affiliations:** 1Department of Pharmacology and Therapeutics, Faculty of Medicine, Collegium Medicum in Bydgoszcz, Nicolaus Copernicus University, 85–090 Bydgoszcz, Poland; michal.wicinski@cm.umk.pl (E.W.); ewelhan@gmail.com (E.M.); ewelina.sawicka@cm.umk.pl (E.S.); bartosz.malinowski@cm.umk.pl (B.M.); 2Department of Geriatrics, Faculty of Health Sciences, Collegium Medicum in Bydgoszcz, Nicolaus Copernicus University, 85–090 Bydgoszcz, Poland; marta.podhorecka@cm.umk.pl; 3Department of Neurosurgery, Neurotraumatology and Paediatric Neurosurgery, Faculty of Medicine, Collegium Medicum in Bydgoszcz, Nicolaus Copernicus University, 85–090 Bydgoszcz, Poland; sniegocki.m@gmail.com; 4Department of Plastic, Reconstructive and Aesthetic Surgery, Faculty of Medicine, Collegium Medicum in Bydgoszcz, Nicolaus Copernicus University, 85–090 Bydgoszcz, Poland; drszychta@gmail.com

**Keywords:** polyphenols, gut microbiota, probiotics, antioxidants

## Abstract

Polyphenols form a diverse group of compounds containing at least two hydroxyl groups in their chemical structure. Because of the common presence in plant kingdom, polyphenols are considered a significant component of food and an important group of compounds with antioxidant properties. The absorption of polyphenols present in food depends mostly on the activity of intestinal microflora. However, little is known about the processes and interactions responsible for such phenomenon in guts ecosystem. There are only few available publications that examine the effect on polyphenols on intestinal microbiota. Therefore, this work will focus on describing the relationship between polyphenol compounds present in food and bacteria colonizing the intestines, their mechanism, and impact on human’s health.

## 1. Introduction

### Intestinal Microflora

In human’s gastrointestinal tract, there are over one thousand species of bacteria. The representatives of eukaryotes and prokaryotes such as *Firmicutes*, *Bacteroidetes*, *Actinobacteria,* and *Proteobacteria* form a permanent or temporary component of microbiota [1]. Among them there are symbiotic and commensal microorganisms, but also those that can cause diseases. Intestinal microflora take part in creating short-chain fatty acids (SCFAs) and branched-chain fatty acids (BCFAs) that are important nutrients for intestinal cells [2]. SCFAs not only stimulate division and differentiation of enterocytes, but also regulate and maintain the mineral balance of organism and promote iron, calcium, and magnesium absorption [3]. The products that are created during fermentation of microorganisms regulate metabolism of lipids and glucose [4]. Microflora of gastrointestinal tract stimulates the synthesis of mucin, which protects organisms from toxic substances and pathogens [5]. The protective function of the intestine is also connected to the degradation of toxins and insulation of its walls. The microorganisms colonizing human’s gut are responsible for modulating the immune system [6] and participate in the production of vitamins such as B and K. Despite of all the above-mentioned benefits, the bacteria could also cause negative changes in health. Abnormal microflora can impact the growth of neoplasm in gastrointestinal tract, mainly in the large intestine due to production of mutagens and carcinogens (*E. coli, E. faecalis, Basteroides*) [7]. Low content of fiber, high content of protein and fats, but also unhealthy, highly processed food can cause disturbance of microbiota [8]. Moreover, taking medications such as antibiotics, chemotherapeutics, non-steroidal anti-inflammatory drugs, proton pump inhibitors, antineoplastic agents, stress, infections, and stimulants might affect microflora in a similar way to incorrect food content [9,10]. The possible effect of such a disturbance is the development of allergy [11] or diabetes mellitus type 1 [12]. Additionally, recent studies show the connection between intestinal microbiota and the central nervous system where the imbalance might be a possible cause of mental diseases e.g., depression [13,14]. Dysbiosis in gastrointestinal tract can also affect metabolic disorders and have an impact on obesity [12], heart diseases, and diabetes mellitus type 2 [15].

## 2. Phenol Compounds 

Phenols represent a very important group of antioxidant compounds, widespread among plants [16] that arise as products of plants’ metabolism in the wake of stress, damage, infection, or UV radiation. They are a cohort of compounds of diversified chemical structure emergent in two pathways. In shikimic acid’s transformation, hydroxycinnamic acids and coumarins come into existence. Simple structure polyphenol compounds are made of acetic acid. Structurally more complex flavonoids are formed as a result of these two routes (Figure 1). [17]. 

Due to the composition, polyphenol compounds might be differentiated into phenolic acids, flavonoids, stilbenes, and lignans (Table 1) [18]. Flavonoids are a large group of compounds composed of two benzene rings connected by a tri-carbon chain or heterocyclic ring [19]. Flavonoids can occur in two forms: Free - aglycons, or in the form of β-glycosides [combination of aglycone with the sugar part] [19]. Aglycones of most flavonoids are more hydrophobic than their glycosides [20]. They can penetrate the biological membranes of enterocytes as a result of passive transport [20]. The presence of a glycosidic substituent in the flavonoid molecule causes an increase in its mass and hydrophilicity, which limits its absorption as a result of diffusion [21]. Flavonoid compounds that have not been absorbed in the upper digestive tract reach the large intestine, where they are modified by bacterial enzymes [22]. These compounds can be metabolized by β-glucosidase, α-rhamnosidase, and β-galactosidase, synthesized by bacteria [23]. The resulting flavonoid aglycons can be absorbed in the large intestine or further metabolized [24]. As a result of metabolism, many low-molecular phenolic compounds, such as phenylacetic and phenylpropanoic acids, are formed [25]. Polyphenols commonly occur in plants; they are a significant ingredient of food. Fruits, especially berries, strawberries, raspberries, and chokeberry are abundant sources of them. Polyphenols occur in a fruit juice as well. Content in these products depends on the stage of manufacture. Juices made of freshly extruded fruits contain distinctly more polyphenols in comparison to manufactured concentrates [26]. Vegetables are a rich source of polyphenol compounds than fruits, mostly phenolic acids. Garlic, onion, broccoli, red cabbage, and red pepper pose the best source of them [27]. Green and black tea are rich in epicatechin, catechin, kaempferol, and quercetin [28]. Polyphenols’ antioxidant properties are connected to the presence of many hydroxyl groups, and thanks to this characteristic they protect plants from free radical actions. There are two mechanisms of polyphenol antioxidant action. Polyphenol reaction with radicals leads to the rise of a phenol radical, which due to a low reactivity does not cause any threat to the cells [29]. Antioxidant properties of polyphenol compounds result from the ability of metal chelation [30]. Transition metals (especially iron and copper) are elements present in many proteins and act out an important role in the functioning of the cells. Those metals are involved in Fenton’s reaction [31], which consists of two stages. The first one is the reaction between metal and hydrogen peroxide forming hydroxyl radical. Afterwards, the reduction of oxidized metal by superoxide anion takes place [32]. The arisen hydroxyl radical is a reactive form of oxygen that reacts with most of the compounds present in the cell e.g., protein, fatty acids, nucleic acids, or small metabolites [33]. Polyphenols inhibit the activity of oxidases, which catalyze the formation of water or hydrogen peroxide [34]. However, in specific situations, they can show prooxidase action via reduction of transition metals that allows catalysis of Fenton’s reaction (Figure 2) [35]. 

## 3. The Influence of Polyphenol Compounds Contained in Tea on Intestinal Microflora.

Tea, one of the most popular drinks in the world, is high in polyphenols such as catechin, epicatechin, and quercetin. Moreover, it is known for its antitumor properties, especially in the case of large intestine cancer. Several studies tried to examine the influence of polyphenol compounds present in tea on bacteria in gastrointestinal tract. Tzounis et al. [38] examined the influence of monomers of flavan-3-ols like (-) epicatechin and (+) catechin on fecal bacterial growth (Table 2). The experiment was carried out in conditions that correspond to those present in the human intestine with the usage of two doses, respectively 150 mg and 1000 mg. Presence of (+) catechin stimulated the growth of *Clostridium coccoides–Eubacterium rectale*, *Bifidobacterium spp.,* and *Escherichia coli*, but at the same time inhibited the growth of *C. histolyticum*. In comparison, (-) epicatechin stimulated only the growth of population of *C. coccoides–Eubacterium rectale.* Due to prebiotic properties, the consumption of food enriched form polyphenols contributes to gut integrity and intestinal homeostasis [39]. Growth of the group *Clostridium coccoides–Eubacterium rectale* is beneficial for the host due to increased production of short-chain fatty acids (SCFAs) used by the cells as a source of energy [40]. SCFAs formed by bacteria also inhibit proliferation of tumor cells and accelerate conversion of cholesterol into bile acids [41]. Growth of *Bifidobacterium spp.* was minor, but statistically significant as those bacteria inhibit proliferation of pathogens by production of organic acids [42]. Moreover, decrease in a population of *C. histolyticum* could be observed. Those bacteria due to the proteolytic properties can redound to the progression of colon cancer and development of inflammatory bowel diseases such as Crohn’s disease or ulcerative colitis. Therefore, decline in the number of *C. histolyticum* brings benefits for the host. The mechanism of this process is connected to degradation of proteins. Undigested proteins undergo the fermentation in the large intestine; the process is carried out by proteolytic bacteria (mainly *Bacteroides* and *Propionibacterium*) during which ammonia, thiols, amines, phenol, and indole compounds are formed. Additionally, *Amines* in the nitrosation reaction can form carcinogens such nitrosamines. Phenol and indole compounds negatively affect intestinal epithelium as they stimulate development of cancer by increasing the strength of carcinogenic factors [43,44]. In contrast to glycosides in food that do not seem to cause any harm in the host organism, flavonoid aglycones present mutagenic and toxic effects. The contradiction in the results of the influence of flavonoid aglycones on human health is connected with the transformation to compounds either working profitably or unfavorably for the host [45].

In Goto et al.’s study [46], the polyphenols present in tea has a positive effect on the growth of *Bifidobacterium spp..* Decline in the number of putrefactive bacteria like *Enterobacteriaceae spp*. and *Clostridium spp.* was noticeable as well. The consumption of tea evidently improved conditions in the intestines by lowering the levels of sulfides, ammonia, and pH.

In Lee et al.’s experiment [47], the main aim was to examine the effect of polyphenols and their aromatic metabolites on bacterial growth. A significant inhibition of the growth of *Clostridium perfringens, Clostridium difficile, E. coli, Salmonella,* and *Staphylococcus spp* was observed. The examined compounds were also inhibited, but in a smaller extent. The growth of probiotic bacteria e.g., *Clostridium spp., Bifidobacterium spp., Lactobacillus spp* was noticed.

## 4. The Influence of Polyphenols Present in Plant Extracts on Microbiota

Plant extracts form a large class of compounds with multi-directional effects on human health. Their properties are the sum of individual functions of complexes included in a specific extract. Commonly, together with polyphenols, other antioxidants like vitamin C, E, and carotene are present. Those substances have protective properties towards different tissues. The pharmaceutic industry offers abundant numbers of products containing distinct plant extracts used as a prevention or treatment in conditions like cardiovascular disease, liver dysfunction, and to mitigate symptoms of menopause.

Yamakoshi et al. [48] studied the effect of grapes’ seeds extract that contained, respectively, 89.3% and 38.5% proanthocyanidins on intestinal bacteria. The beverage was administered to two groups for two weeks. The number of *Bifidobacterium* increased when at the same time, the colony of *Enterobacteriaceae* decreased. *Bifidobacterium* as bacteria exhibit positive effects on human health through the formation of SCFAs, but also by lowering the pH and inhibiting the development of putrefactive bacteria.

Since intestinal microflora is very sensitive and easily detects changes in external conditions, it is of high importance to supplement with substances such as prebiotics (non-absorbable compounds) that stimulate growth of one or few groups of beneficial bacteria in colon [49].

Tzounis et al. [50] evaluated the influence of flavonols derived from cocoa with the potential of being used as prebiotics. Patients were divided into two groups where the first one obtained 494 mg of cocoa flavonols per day and the other group was given 23 mg of the same flavonols daily for the period of four weeks. Researchers determined a statistically significant growth of *Bifidobacterium* and *Lactobacillus*. Both types of bacteria inhibit development of pathogenic microorganisms. Moreover, studied extract affected the population of *C. histolyticum* by relevantly decreasing it. Cocoa-derived flavonols can also inhibit the growth of *Clostridium perfringens* that might contribute to progression of colon cancer and participate in arising of inflammatory bowel disease. The change in intestinal microflora was connected to a change in the level of triglycerides in patients’ blood. Therefore, the study confirms the possibility of applying the polyphenols as a potential prebiotic compound.

## 5. The Effect of Polyphenols Contained in Wine on Intestinal Microflora

Red wine is a source of multiple polyphenol compounds e.g., catechin, proanthocyanidins, stilbenes, anthocyanins, and flavonols that exhibit cardioprotective and antioxidant properties. Despite many studies on the mechanism of action of red wine, the contribution of individual components was not unequivocally determined. Compounds that potentially have the greatest meaning are resveratrol, phenolic acids, and proanthocyanidins. The positive effect of red wine on the gut microbiota is due to prebiotic activity and inhibition of pathogen growth [51].

Queipo-Ortuno et al. [52] analyzed the effect of consumption of red wine on intestinal microbiota. For four weeks, they studied the influence of wine (272 mL), non-alcoholic wine (272 mL), and alcohol (250 mL). There was a significant growth of *Enterococcus, Prevotella, Bifidobacterium, Bacteroides uniformis, Eggerthella lenta,* and group *Blautia coccoides–Eubacterium rectale* in patients receiving red wine. Simultaneously, during this study, blood pressure measurements and lipid concentration tests were carried out on patients. The obtained results showed some beneficial effects of polyphenols present in wine on circulatory system and lipid profile. Not only systolic and diastolic blood pressure, but also the concentration of triglycerides, total cholesterol, and HDL cholesterol significantly declined. Those changes could be associated with the growth of a species *Bifidobacterium spp*. Bifidobacteria can produce short-chain fatty acids capable of reducing cholesterol synthesis in the liver or can bind bile acids in the digestive tract [53]. The obtained results present some potential advantages of the usage of red wine polyphenols as a prebiotic source. The experiment did not detect any effect of wine or alcohol on the growth of *Lactobacillus spp*., while consumption of alcohol ensued from increase in proliferation of *Bacteroides spp*. and *Clostridium spp.* Additionally, the disappearance of *Prevotellaceae spp.* population in comparison with red wine and non-alcoholic wine occurred. The concentration of uric acid considerably decreased when patients received red wine possibly as a result of the growth of *Proteobacteria* that degrade the mentioned compound [54]. Lowering of CRP concentration after non-alcoholic and red wine period might be connected to the increase in the number of *Bifidobacterium spp.,* and the change in the recalled parameter can indicate an advantageous effect of wine polyphenols on cardiovascular system [55].

A similar study was carried by Moreno-Indias et al. [56] where researchers examined prebiotic effect of red wine on intestinal microbiota and reduction of metabolic syndrome markers in obese patients. In this study, red wine (272 mL) and non-alcoholic wine (272 mL) were tested. After the period of red and non-alcoholic wine consumption, in patients with a metabolic syndrome, scientists observed a relevant increase in the number of *Fusobacterium* and *Bacteroidetes* while at the same time, a significant decline in the population of *Firmicutes* was noticed. Likewise, in the group of healthy subjects, a statistically important growth of *Bacteroidetes* in comparison with the initial value was noticeable. In the case of patients with dysmetabolic syndrome X, during red as well as non-alcoholic wine consumption, researchers registered decline in *Clostridium* and *Clostridium histolyticum* species that are within *Firmicutes* phylum. At the same time, a growth of bacteria in the group of *Blautia coccoides–Eubacterium rectale*, *Faecalibacterium prausnitzii, Roseburia,* and *Lactobacillus* was noticed. In healthy patients, scientists observed an increase in the number of *Faecalibacterium prausnitzii* and *Roseburia.*

In the study conducted on obese women and men, a significant decrease in the number of Bacteroidetes and a tendency to reduce the number of *Faecalibacterium prausnitzii* in men was noted. These changes were not observed in women. The differences may be due to hormonal changes [57]. *Faecalibacterium prausnitzii* belongs to the group of important SCFA producers [58]. A decrease in the number of these bacteria is observed in patients with inflammatory bowel disease [IBD] [59]. The most common diseases from this group are Leśniowski-Crohn’ s disease and ulcerative colitis [59]. In IBD, the excessive activation of the immune system is increased by pathogenic bacteria [60]. Additionally, SCFA-inducing interleukin 10 [IL-10] reduces inflammatory reactions [60]. *Akkermansia muciniphila* is an important factor modulating *F. prausnitzii* count. *A. muciniphila* degrades mucus components to propionic acid, acetic acid, and oligosaccharides, which are used by *F. prausnitzii* [61]. Numerous clinical studies have investigated the benefits of polyphenol supplementation (anthocyanins, epigallocatechin gallate, resveratrol, and curcumin) in patients with IBD. The improvement of patients’ health resulted from multidirectional actions such as regulation of cytokine concentration, antioxidant enzymes, and modification of intestinal microflora [62,63]. Anthocyanins are natural pigments widespread in plant kingdom and are present in flowers, fruit peel, or in wine where they impart the color. Hidalgo et al. [64] determined the influence of mixture of anthocyanins, 3-glucoside of malvidin (oenin), and gallic acid on bacterial growth. The use of oenin on microorganisms caused the increase in the total number of bacteria. Scientists demonstrated the growth in population of *C. coccoides–Eubacterium rectale* that is responsible for production of SCFA. The usage of mixture of anthocyanins of low concentration on studied bacteria had a positive correlation with the growth of *Bifidobacterium spp.* and *Lactobacillus spp.* but did not affect *C. coccoides-Eubacterium rectale* and *C. histolyticum* populations. Gallic acid used in the experiment inhibited the growth of *C. histolyticum* without any negative consequences on the beneficial bacteria population in intestines. Moreover, it considerably increased the total number of bacteria and specific species called *Atopobium spp.,* which is significant in gastrointestinal tract as *Atopobium minutum* induces the process of apoptosis of colon cancer cells. Therefore, the growth of this bacteria suggests a positive effect on human health [65]. Anthocyanins and their metabolites present in large intestine, increase the number of *Lactobacillus spp.* and *Bifidobacterium spp.* These bacteria have an ability to metabolize phenol complexes during augmentation, supplying energy to the cells and enriching the environment for other bacteria. Hence, anthocyanins, mainly the mixture of anthocyanins derived from grapes, might function as a stimulator for growth of *Lactobacillus spp.* and *Bifidobacterium spp.* that have a positive effect on large intestine. The increase in number of lactic acid bacteria in colon reduces the creation of procarcinogens in the large intestine, and lowers pH and cholesterol levels in patients (Figure 3).

## 6. Polyphenols’ Negative Impact on Microflora of Intestines

Polyphenols act positively on a composition of intestines’ microflora and have a beneficial impact on human health. The impact of polyphenols on bacterial growth depends on the dose, structure, as well as phylum of bacteria [66]. Gram-negative are more resistant to polyphenols than gram-positive due to the differences in the cell wall structure [67]. The antibacterial effect is connected with various mechanisms, for instance cell wall damaging [68], H_2_O_2_ production [69], or changes in permeability of cell membrane [70]. Polyphenol compounds might derange quorum sensing [71]. Another hypothesis envisages the creation of metal-polyphenol compounds. The lack of iron suppresses the growth of sensitive bacteria, mostly aerobic microbes [72]. In Firmann et al.’s study [73] the influence of quercetin on intestinal commensal bacteria such as *Ruminococcus gauvreauii*, *Bifidobacterium catenulatum* and *Enterococcus caccae*, was evaluated. Outcome of studies show that quercetin does not influence the growth of *Ruminococcus gauvreauii* but mildly hinder the growth of *Bifidobacterium catenulatum* and *Enterococcus caccae*. The study held by Duda-Chodak [74] ascertained negative effect of aglycons (naringenin, hesperetin, quercetin, and catechin) and glycosides (naringin, hesperidin, and rutin) on intestinal microbiota represented by *Bacteroides galacturonicus, Lactobacillus spp., Enterococcus caccae, Bifidobacterium catenulatum, Ruminococcus gauvreauii,* and *Escherichia coli*. In this experiment, polyphenols in concentrations of 20, 100, and 250 μg/mL diluted in growth medium were used(for quercetin the concentrations were 4, 20, and 50 μg/mL). The examined aglycones presented inhibiting properties towards investigated bacteria (dependent on the concentration of the substance in microbiological medium) while glycosides did not exhibit any effect on intestinal microflora. Catechins did not affect the growth of microorganisms. Only the growth of *B. catenulatum* was slightly slowed down while the growth of *E. caccae* was stimulated by higher concentrations of catechins.

The study carried by Tabasco et al. [75] evaluated the influence of grapes’ polyphenols on the growth of lactic acid bacteria and *Bifidobacterium* with the usage of three extracts of concentrations 0.25, 0.5, and 1 mg/mL, respectively. The experiment showed a dose-dependent inhibition on growth of above-mentioned bacteria.

Puupponen-Pimia et al. [76] determined the effect of pure polyphenols and polyphenols contained in the extract from blueberries on the probiotic bacteria, *Escherichia coli* and *Salmonella*. Lactic acid bacteria were more resistant to pure polyphenol compounds than other examined microorganisms, except myricetin, which inhibited the growth of all lactic acid bacteria with no effect on the growth of *Salmonella*. Phenolic acids in high concentration mainly inhibited the growth of Gram-negative bacteria. The extracts showed the growth-inhibiting effect, mostly towards Gram-negative bacteria These results indicate that the inhibitory effect of the extracts is due to the synergistic action of polyphenols.

**Table 2 nutrients-12-00350-t002:** The effects of the intake of polyphenols on intestinal bacteria.

Examined Polyphenols	Dose	Duration	Type of Experiment	Subjects	Diet	Diseases	Medications Taken	Stimulation of Bacterial Growth	Inhibition of Bacterial Growth	Reference
(+) – catechin (-) - epicatechin	150 mg/1000 mg	48 h	In vitro					Escherichia coli Bifidobacterium spp. Group Clostridium coccoides- Eubacterium rectale	Clostridium histolyticum	[38]
Tea catechins	300 mg	6 weeks	In vivo [Open label pilot]	31 females and 4 males from 66 to 98 years of age	38.7 g proteins, 21.1 g lipids, 191 g carbohydrates, minerals, and vitamins.	Hypertension, cerebral stroke, senile dementia	None antibiotics	Bifidobacterium Lactobacillus		[46]
(+) – epicatechin (-) – catechin 3-O-methylgallic acid Gallic acid Caffeic acid Phloretic acid 3-Phenylpropionic acid 4-Hydroxyphenylacetic acid	1 mg/mL	24 h	In vitro					Clostridium spp., Bifidobacterium spp., Lactobacillus spp.	Clostridium perfringens, Clostridium difficile, Escherichia coli, Salmonella, Staphylococcus spp..	[47]
Extract from grapes’ seeds containing proanthocyanidins	0,19 g 0,38 g	2 weeks	In vivo [Open label pilot]	9 healthy adults from 37 to 42 years of age 8 elderly inpatients from 67 to 98 years of age	None red wine, green tea, products prepared using lactic bacteria 62,5±4,7 g/d proteins, 38,0 ± 4,7 g/d lipids, 212,0 ± 15,0 g/d carbohydrate 9,4 ± 1,3 g/d sodium, 803,9 ± 105,8 mg/d moisture, 568,1 ± 89,9 mg/d calcium	Cerebral stroke, bone fracture, senile dementia, articular rheumatism	None antibiotics None antibiotics	Bifidobacterium spp.	Enterobacteriaceae	[48]
Cocoa flavonoids	23 mg 494 mg	4 weeks	In vivo (Randomized, double-blind, placebo- controlled)	22 healthy volunteers (12 male and 10 female - premenopausal ) from 18 to 50 years of age	The patients did not change their dietary habits		None antibiotics	Bifidobacterium spp. Lactobacillus spp.	Clostridium spp.	[50]
Polyphenols in wine	272 mL/d	4 weeks	In vivo (Randomized, crossover, controlled, intervention)	10 healthy adult men from 45–50 years of age	The patients did not change their dietary habits None alcohol and red wine		None antibiotics and supplements	Enterococcus, Prevotella, Bacteroides, Bifidobacterium, Bacteroides uniformis, Eggerthella lenta, Blautia coccoides–Eubacterium rectale group		[52]
Polyphenols in wine	272 mL/d	30 days	In vivo (Randomized, crossover, controlled intervention)	20 adult men (10 obese and 10 healthy) from 45 to 50 years old:	The American Heart Association dietary guidelines		None antibiotics, prebiotic, probiotics, symbiotics, vitamin supplements and any medical treatment influencing intestinal microbiota	Bifidobacterium spp. Lactobacillus spp.	Enterobacter cloacae Escherichia coli	[56]
Resveratrol Epigallocatechin-3-gallate	80 mg/d 282 mg/d	12 week	In vivo (Randomized, double-blind, placebo- controlled)	18 males and 19 females from 20 to 50 years of age	< 600 mg caffeine, < 3 cups green tea, < 20 g alcohol	Overweight and obese	None antibiotics, medication, supplements		Bacteroidetes	[57]
Anthocyanins Gallic acid		24 h	In vitro					Bifidobacterium spp. Lactobacillus spp.	Clostridium histolyticum	[64]
Quercetin	12.5, 25, 50, 75μg/mL	24 h	In vitro						Bifidobacterium catenulatum, Enterococcus caccae	[73]
Naringenin, Naringin, Hesperetin, Hesperidin, Rutin, Quercetin Catechin	20, 100, 250 μg/mL* * [for quercetin 4, 20, 50 μg/mL]	24 h	In vitro						Lactic acid bacteria	[74]
Grapes’ polyphenols	0.25, 0.5, 1 mg/mL	24 h	In vitro						Bifidobacterium spp. Lactobacillus spp.	[75]
Extracts from: blackcurrant blueberry cranberry cloudberry lingonberry raspberry berry of sea-buckthorn strawberry Polyphenols: Apigenin, Caffeic acid, (+) - Catechin, Chlorogenic acid, Coumarin-3-carboxylic acid, Cyanidin chloride, Delphinidin chloride, Ferulic acid, Isoquercetin, Kaempferol, Cyanidin-3-O-glucoside, Luteolin, Myricetin, Pelargonidin chloride, Quercetin dihydrate, Rutin trihydrate, Trans-cinnamic acid	0,5, 1, 5 mg/mL	24 h	In vitro						Lactobacillus spp.	[76]

## 7. Conclusions

Reviewed studies present complex interactions between polyphenol compounds and intestinal microorganism with both positive and negative consequences. The presented studies examined the influence of polyphenols on gut’s microbiota in vivo and in vitro. Clinical studies, due to their design, do not give unequivocal answers. The first major limitation of the research is the number of individuals. A study group that is too small may not be enough to notice subtle changes. Another limitation is the short duration of the test. Making a comparison of the results between studies is often difficult due to differences in the accuracy of microorganism identification; some studies indicate changes at the phylum level, and others at the level of the species. In addition, the lack of clear information about the diet and lifestyle of patients means that caution should be exercised when extrapolating the results obtained. The in vitro studies were performed only on monobacterial cultures. As previously mentioned, intestinal microbiome is one of the most complex ecosystems on Earth. Biotransformation of polyphenol complexes by bacteria is still a subject of discussion and requires further studies. In respective stages, different microorganisms are responsible for transformation of polyphenols. Therefore, a lack of knowledge on this subject limits the analysis of potential benefits for human. Due to the limitations of the above-mentioned studies, further experiments are essential to examine the contribution of particular bacteria on different stages of biotransformation of polyphenol compounds and the consequences for human’s health.

## Figures and Tables

**Figure 1 nutrients-12-00350-f001:**
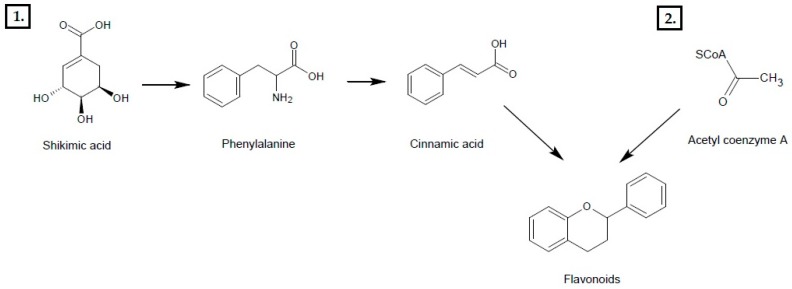
Biosynthesis of Flavonoids. 1. The shikimic acid pathway; 2. The acetate pathway.

**Figure 2 nutrients-12-00350-f002:**
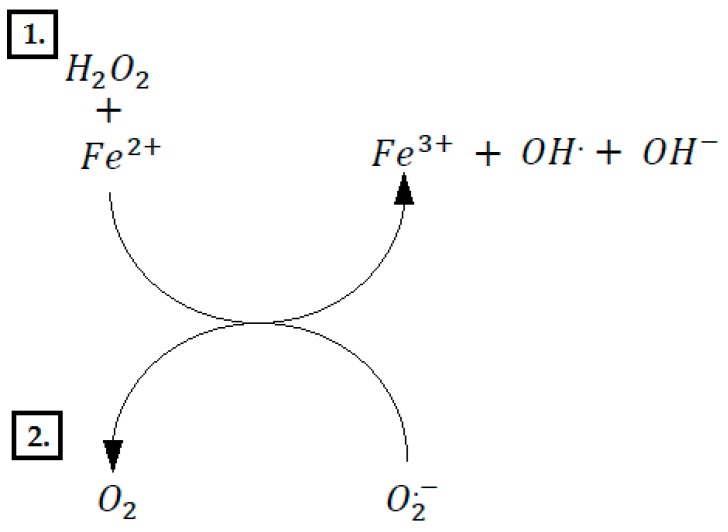
Fenton’s cycle. 1. $H_{2}O_{2}$ is broken down into $OH^{.}$ and $OH^{-}$ in the presence of $Fe^{2+}$ 2. Reduction of $Fe^{3+}$ to $Fe^{2+}$ by superoxide anion radical.

**Figure 3 nutrients-12-00350-f003:**
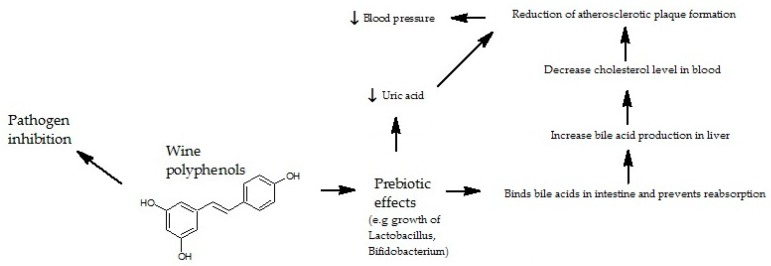
Interaction between wine polyphenols and gut microbiota.

**Table 1 nutrients-12-00350-t001:** The classification of natural polyphenols [36,37]

Class	Subclass	Examples of Compounds	Source
Phenolic acids	Hydroxycinnamic acids	Curcumin Caffeic acid Ferulic acid	Fruit and cereals
	Hydroxybenzoic acids	Gallic acid Protocatechuic acid Vanillic acid	Onion, raspberry, blackberry, strawberry
Favonoids	Flavonols	Kaempferol, Quercetin, Myricetin	Onions, tea, lettuce, broccoli, apples
	Flavanones	Naringenin, Hesperetin	Oranges, grapefruits
	Flavanols	Gallocatechin Catechins	Tea, red wine, chocolate
	Isoflavones	Genistein, Glycitein, Daidzein	Soybeans, legumes
	Anthocyanins	Pelargonidin, Delphinidin, Malvidin	Blackcurrant, strawberries, red wine, chokeberry
	Flavones	Apigenin, Luteolin,	Parsley, celery, red pepper, lemon, thyme
Stilbenes		Resvertrol	Red wine
Lignans		Pinoresinol, Lariciresinol, Secoisolariciresinol, Sesamin	Flax seed, sesame seed, red wine

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
