# Peer review of "The Influence of Polyphenol Compounds on Human Gastrointestinal Tract Microbiota"

_nutrients, 2020, doi:10.3390/nu12020350_

Round 1

Reviewer 1 Report

Polyphenols are understood as molecules of plant origin with positive effects in warm blooded animals including humans. This review then focus on the interactions of polyphenols with gut microbiota. While the topic is interesting, the review is, unfortunately, confusingly arranged. This is a type of a review in which authors took certain number of published data, extracted a few sentences or ideas from each published paper but did not include their own opinion, what is likely correct and what is less likely. In a consequence, reader cannot learn too much since conflicting sentences or paragraphs are placed one after another. In addition, quality of language is not too. Inappropriate terms are used and sentences are complicated. Authors also use different terms to describe the same compound what makes the text impossible to follow. The whole manuscript can be corrected. But this would represent an extensive re-editing and re-writing.

Specific comments

It is not mentioned in the whole manuscript that the review deals with humans. It could be mice, pig, chickens, cattle etc. Please, correct.

line 28, intestinal tract is not the richest niche with microbial colonisation. You may say dense, highly populated etc. But in terms of richness, i.e. numbers of different species, river water r soil microbial populations are much more diverse.

l.30, not at all, there are no two groups, this is an extreme human bias. There is only gut microbiota. Humans can be positive for Clostridium perfringens, Cl. difficille, Salmonella etc without any clinical signs and despite this, you would classify these bacterial species as pathogens. E. coli is mostly commensal but clones with a few specific genes may represent real pathogens. There is a real gradient of interactions between host and microbiota which must be simplified for undergraduate students but not in scientific literature.

35 and 36, add references at the end of the sentences

l.38, why intestines, why plural. Do humans have more intestines?

l.41, advantage? What advantage? Reword.

l.43, please specify which microbiota members produce which mutagens and carcinogens and support this with references.

l.54, polyphenols are not particles

l.57-59, you define two pathways but these are not clearly explained. In fact, it seems that there are three possibilities, shikimic acid transformation, acetic acid transformation, and third pathway leading to the production of complex phenolic compounds employing both shikimate and acetate pathways.

65, vegetable do not encapsulate, please reword 66, Garlic, onion, broccoli, red cabbage, red pepper do not pose, please reword

l.67, Green and black tea do not comprise, please reword

l.69-72, you mix up two ways of polyphenol antioxidant activity, i.e. phenol radical formation and metal chelation. But you have to say clearly in the first sentence that there are two mechanisms of polyphenol antioxidant action...

l.91, Obtained data suggest that... delete this, you commonly use these accessory sentences but these are useless.

l.92, “consumption of food ... helps maintaining healthy intestines thanks to prebiotic properties”, this sentence is somewhat naive. It should say “Due to prebiotic properties of ... consumption of food enriched form polyphenols contributes to gut integrity and intestinal homeostasis.

l.103, genus Clostridium is so broad that it is of a questionable value to use genus name without any other specification

l.104 and 105, amines are mentioned twice, in two sentence. Why, this is confusing.

108, glucosides are not delivered, these are present in food

l.108 and 109, without any introduction, two new terms suddenly appeared, glucosides and aglycones.

l.113, in a form of growth, please reword.

117, delete “Scientists noticed” and reword 128, respectively should be at the end of the sentence

l.130, After appointed time the observations were made and scientists stated that – delete all of this. You commonly use this style but this does not bring any information, this is just a plain volume which only complicates extraction of information.

l.134-136, why just this paragraph formed by a single sentence?

l.155, After experiment’s termination the scientists observed – as above, do not use such style, delete this intro semi-sentences.

l.156, when you specify genus Bacteroides, this covers also the species Bacteroides fragilis.

l.160-162, why do you think that blood biochemistry is influenced by Bifidobacterium?

164, perhaps low case “w” in With

l.189, why butanoate when in the rest of the manuscript you use a broader term SCFA

l.189, this is a terrible mistake – How can you write about insignificant changes!!! This is absolutely impossible. There are many insignificant changes in any experiment....

l.202, Scientists alleged -  as above, delete this redundant and useless words

l.206, Presented articles show that - as above, delete this redundant and useless words

l.207, Phenol compounds might exhibit antimicrobial activity. This sentence does not tell anything. But when readers read a review they want to learn something

l.218, what is negative significance? I can understand negative effect but not significance.

l.221, 223, 224 and 225, typical confusing presentation characteristic for this review. In line 221, compounds are specified by their names. But later on, without any explanation, authors use terms aglycones, glycosides, catechins... This is not allowed at all.

l.234-235, another example of loose style of this manuscript. First sentences says “no effect on Salmonella”. But just the following sentences says that there was growth-inhibiting effect towards Gram negative bacteria – but Salmonella in previous sentence is Gram negative bacterium. And then follows a sentence that “lactic acid bacteria were more resistant to pure polyphenol compounds. Well, but the first sentence in this triad says that lactic acid bacteria were inhibited by myricetin. So what is right, what is wrong? There is conflicting information in 3 successive sentences and authors do not mind this. How can readers deduce what is correct and what is false. Or that you now play with words and differentiate between pure compounds like myricetin and “extracts”, i.e. complex products.

Table 1. I did not find any reference to the table in the text. Remove column “Author” and rename column “Bibliography” to “Reference”.

All the references are numbered twice.

Author Response

Dear Reviewer,

We greatly appreciate your thoughtful comments that helped improve the manuscript.  All changes are marked in the manuscript.

Thank you very much for your effort.

Sincerely,

Jakub Gębalski

Department of Pharmacology and Therapeutics,

Faculty of Medicine, Collegium Medicum in Bydgoszcz,

Nicolaus Copernicus University,

85-090 Bydgoszcz,

Poland

Reviewer 2 Report

The authors WiciÅ„ski et al. have reviewed the effects of polyphenols on intestinal microbiota. Authors utilized current literature to form the manuscript. However, there are quite a few comments which needs to be addressed before publication. 

<Abstract>

Lines 14-15: "Because....antioxidant properties" can be improved, the word organisms is vague. 

<Effect of polyphenols in wine and overall comments>

How exactly does wine help on microbiome? Although authors have mentioned conclusions based on the various studies, perhaps a mechanism of illustration would be beneficial. 

More descriptive or studies involving with polyphenols on health and disease can be added. For examples, most of the studies mentioned in the table are in vivo or invitro, but not clinical studies. Are there significant number of studies, to be discussed in this review to make it more appropriate to the readers. 

A list of most the polyphenol compounds (Table) can be beneficial with their classifications/source. 

Stability of polyphenols can also be briefly discussed? Whether polyphenols have effect on diseased, healthy, or have prebiotic effect as they seem to modulate the microbiome.

<Table>

The entire table format seems to be broad. Authors can improve the format by adding more information on what are the study types? For examples, authors mentioned in vivo, in vitro but however, what kind of studies they are diseased models, or normal healthy or any other. And in what dosage forms? was there any differences in food delivery or oral or others/? 

<References>

All the references are not recent, authors should try to add the latest references, thus more recent studies.

<Figure>

Authors should come up with an innovative illustration or two or figure based on the title of the manuscript, can be beneficial. 

Author Response

(The authors gave the same response as above.)

Round 2

Reviewer 1 Report

In my original comments I wrote to editor that your review is technically correct but you only, perhaps mainly, combined published data, and you did not enrich them for your opinions, inputs. This is possible but I like more challenging reviews, in which authors express their own opinions and help readers to orient themselves, even in a sense that this and this has been published but we believe that this is not correct because of this and that. And exactly this way of presentation, challenging, provocative and stimulating ideas I missed in your review. Of course, when going for such a challenging review, you have to know, you have to perform many experiments by yourselves alone, based on this you have to be self-confident with you your own opinion. For the moment, it is not so important but keep this in your mind for a future.

Author Response

Dear Reviewer,

Thank you for your time considering our manuscript „The influence of polyphenol compounds on human gastrointestinal tract microbiota”

Changes to the manuscript:

the figures and legends have been corrected (line 65, 105 and 261).

the Conclusions have been improved and extender (line 309-316)

Thank you for giving me advice about review. I can say with certainty that I will use them in the future.

Yours sincerely,

Jakub Gębalski

Department of Pharmacology and Therapeutics,

Faculty of Medicine, Collegium Medicum in Bydgoszcz,

Nicolaus Copernicus University,

Poland

Reviewer 2 Report

The authors Wicinski et al. reviewed the literature in a revised version on the relationship between polyphenols in food and bacteria colonizing in the human intestine. Although the Authors have put a significant amount of effort to revision the initial manuscript, there are still areas to be addressed. Following are a few comments before publication.

Figures and Tables legends can be more descriptive Table 1 lacks references The conclusion can still be more effective and improved

Author Response

Dear Reviewer,

Thank you for your time considering our manuscript „The influence of polyphenol compounds on human gastrointestinal tract microbiota”

We have revised the paper according to the comments. The responses to comments are attached as below.

Line 65,105 and 261 the figures and legends have been corrected.

Line 309-316 the Conclusions have been improved and extended

Once again, many thanks for your suggestions.

Yours sincerely,

Jakub Gębalski

Department of Pharmacology and Therapeutics,

Faculty of Medicine, Collegium Medicum in Bydgoszcz,

Nicolaus Copernicus University,

Poland
